# Immunomodulatory Compounds from the Sea: From the Origins to a Modern Marine Pharmacopoeia

**DOI:** 10.3390/md22070304

**Published:** 2024-06-28

**Authors:** Edoardo Andrea Cutolo, Rosanna Campitiello, Roberto Caferri, Vittorio Flavio Pagliuca, Jian Li, Spiros Nicolas Agathos, Maurizio Cutolo

**Affiliations:** 1Laboratory of Photosynthesis and Bioenergy, Department of Biotechnology, University of Verona, Strada le Grazie 15, 37134 Verona, Italy; 2Laboratory of Experimental Rheumatology and Academic, Division of Clinical Rheumatology, Department of Internal Medicine, University of Genoa, 16132 Genoa, Italy; 3IRCCS Ospedale Policlinico San Martino, 16132 Genoa, Italy; 4Qingdao Innovation and Development Base, Harbin Engineering University, No. 1777 Sansha Road, Qingdao 150001, China; jian.li@hrbeu.edu.cn (J.L.); spiros.agathos@hrbeu.edu.cn (S.N.A.); 5Bioengineering Laboratory, Earth and Life Institute, Catholic University of Louvain, B-1348 Louvain-la-Neuve, Belgium

**Keywords:** bioprospecting, inflammation, autoimmunity, synthetic biology, drug discovery, genetic engineering, immunomodulation, deep sea, systemic sclerosis, rheumatoid arthritis

## Abstract

From sea shores to the abysses of the deep ocean, marine ecosystems have provided humanity with valuable medicinal resources. The use of marine organisms is discussed in ancient pharmacopoeias of different times and geographic regions and is still deeply rooted in traditional medicine. Thanks to present-day, large-scale bioprospecting and rigorous screening for bioactive metabolites, the ocean is coming back as an untapped resource of natural compounds with therapeutic potential. This renewed interest in marine drugs is propelled by a burgeoning research field investigating the molecular mechanisms by which newly identified compounds intervene in the pathophysiology of human diseases. Of great clinical relevance are molecules endowed with anti-inflammatory and immunomodulatory properties with emerging applications in the management of chronic inflammatory disorders, autoimmune diseases, and cancer. Here, we review the historical development of marine pharmacology in the Eastern and Western worlds and describe the status of marine drug discovery. Finally, we discuss the importance of conducting sustainable exploitation of marine resources through biotechnology.

## 1. Why Does the Sea Matter for Human Health?

From the birth of human civilization to the rise of the modern global economy, the ocean has been a core element for development, providing waterways for exploration, cultural exchanges, and trade [1]. From the philosophers of the classical antiquity to present-day oceanographic expeditions, the scientific study of the sea has been a constant human endeavour stimulated by the fascination with its biological diversity [2].

The classification of marine life, however, is still far from offering a conclusive picture. According to initial estimates, of the ~8.75 million species inhabiting the planet, ~2.2 million live in the oceans, ~91% of which still await description [3]. Recent reconsiderations, however, suggest that marine biodiversity, particularly of fungi, protists, and prokaryotes, has been significantly underestimated, now projected to the millions [4,5].

The latest census of marine life, the World Register of Marine Species, contains ~242,000 species. Despite growing quickly (on average, 2332 new species every year), this repository is expected to include the remaining 1–2 million undescribed species, thus entirely covering marine life, only several hundred years from now [6].

Oceans are still largely underexplored sources of lead compounds [7]. From coral reefs to hydrothermal vents, the ocean is characterized by diverse habitats, including extreme environments such as the deep-sea benthic zones [8], where unique ecosystems thrive. Therefore, the sea is an untapped source of chemodiversity to look for secondary metabolites and bioactive compounds with potential applications in human health, some having already entered clinical practice [9,10,11,12,13]. Immune-mediated inflammatory diseases are a significant burden for national healthcare systems, especially in high-income countries, with an increasing incidence registered over the last three decades [14,15]. Moreover, in nearly all forms of cancer, chronic inflammation is involved in disease development [16]. Pharmacological immunomodulation is, therefore, crucial to restore the homeostasis of the immune system in situations of both over- and under-reaction [17,18].

Virtually all marine phyla, from phytoplankton to invertebrates, produce bioactive compounds with pharmacological potential [19,20,21,22], including anti-inflammatory and immunomodulatory properties [23,24,25,26,27]. Although marine pharmacology has its roots in antiquity, the ocean is witnessing a scientific renaissance propelled by interdisciplinary drug discovery research assisted by powerful, high-throughput technologies like untargeted metagenomics and metabolomics [28,29,30,31].

## 2. Marine Pharmacology in the Mists of Time

From time immemorial, marine flora and fauna have been used in folk and traditional medicine (TM) [32]. However, in most cases, this knowledge is orally transmitted; therefore, ethnomedicinal approaches of drug discovery are not always straightforward. Nonetheless, since TM is the mainstay of healthcare delivery in 80% of African and Asian countries [33,34], this heritage is a valuable resource for the identification of new bioactive compounds [35,36].

Since the dawn of mankind, seaweeds (macroalgae) have been consumed in the diet and for medicinal purposes [37], as revealed by archaeological records from 2500 BC suggesting the trading of kelp (Laminariales) between coastal and mountainous areas of the Peruvian and Chilean Andes [38,39,40]. Similarly, the Incas from the Andean lakes of Peru and the Aztecs in the Valley of Mexico consumed the cyanobacterial species *Nostoc* spp., *Phormidium tenue*, and *Chroococcus turgidus* [41,42].

Marine zootherapy is still practiced in West Africa, Central and South America, and East Asia [43,44,45,46,47,48]. Recent ethnopharmacological studies described at least 300 TM uses of marine animals globally [49,50], of which, however, only invertebrates were confirmed sources of therapeutic compounds [51]. The overexploitation of marine animals in TM, particularly of mammals, hawksbill sea turtles (*Eretmochelys imbricata*), manta rays, and devil rays (Mobulidae), is a recognized risk factor for wildlife population decline and species extinction [52,53,54], and, in turn, precludes the identification of new, potentially therapeutic compounds [55]. Therefore, international regulatory policies are urged to restrict the hunting and marketing of threatened species and to promote the conservation of fragile marine ecosystems [56,57].

## 3. The Origins of Marine Pharmacology and Immunology in the West

Descriptions of medical uses of marine animals are found in Greek and Byzantine texts from the classical antiquity (fifth century BC–7th century AD) [58,59], including the *De Materia Medica* by the father of Western pharmacognosy, Pedanius Dioscorides (40–90 AD) [60], and the *De natura animalium* by the Roman author Claudius Aelianus (175–235) [61]. Similarly, seaweeds are mentioned in the *Corpus Hippocraticum* by Hippocrates of Kos (460–370 BC), in the *Historia Plantarum* by Theophrastus (350–287 BC), and in the *Naturalis Historia* by Gaius Plinius Secundus (Pliny the Elder, 23–79 AD) [62,63]. Moreover, around this time, the Roman author Aulus Cornelius Celsus (25 BC–50 AD) formulated the oldest recorded definition of inflammation: “*Notae vero inflammationis sunt quatuor: rubor et tumor cum calore et dolore*” (the signs of inflammation are four: redness, swelling, fever, and pain) [64]. From the second century BC onwards, technology and science [65,66] constantly flowed between the Far East and the Mediterranean via the Silk Road (*Sichou zhi lu*, 丝绸之路), a network of land and sea trading posts connecting the Greco-Roman world with Mongolia and China via the Middle East, Eurasia, Persia, and India. Besides primarily serving geopolitical interests [67], the Silk Road promoted the reciprocal exhange of medical and pharmacological knowledge between the Far East and the Western world [68,69].

During the Age of Discovery in the fifteenth century, European countries assembled vast collections of flora from overseas, de facto establishing global bioprospecting, although ethnobotanical knowledge was also lost due to the forced conversion of the indigenous peoples to Christianity, especially by the Conquistadores [70]. Nonetheless, exotic species incessantly flowed from the overseas colonies in the four corners of the world to the main cultural centers of Europe, resulting in new discoveries in pharmacology [71] and Western medicine eventually being united in 1948 under the aegis of the International Pharmacopoeia of the World Health Organization [72].

In the 1960s, Western science achieved a deeper understanding of immunity and inflammation with the elucidation of the structure of antibodies and their generation via genetic recombination, as well as the identification of antibody-producing B cells, regulatory T-lymphocytes, and dendritic cells as antigen-presenting cells. Ironically, the concept of autoimmunity—the condition whereby toxic autoantibodies recognise self-antigens, causing chronic inflammation—was formulated in 1892 by German physician Paul Ehrlich, although it was rejected as “physiologically inconceivable” and referred to as “*horror autotoxicus*” [73]. Only in 1965 was autoimmunity recognized as a common immunological disorder underlying the pathogenesis of chronic inflammatory diseases [74]. Eventually, the invention of monoclonal antibodies and their application in clinical practice, as well as the discovery of cellular checkpoint control, paved the way for cancer immunotherapy and targeted therapies for autoimmune diseases [75]. Today, a plethora of inflammatory mediators are known, including sub-populations of immune cell types, their released soluble factors (cytokines, antibodies), and the intracellular genetic and molecular mechanisms which sustain inflammatory disease [76].

## 4. The Evolution of Marine Materia Medica in the Far East

Written records of marine flora appear in the 2500-year-old Chinese book the *Classic of Poetry* (*Shijing*, 诗经) [77,78,79,80]. The main pharmacological heritage in the Far East, however, is the *Bencao*, a series of compendia of materia medica produced over 2000 years [81]. Among the earliest versions, the *Xinxiu Bencao* (新修本草, *Newly Revised Materia Medica*) is the first pharmacopoeia commissioned by Imperial order during the Tang dinasty (618–907) and the oldest example of national codex, describing 850 medicinals, many of which are still used in Chinese TM [82]. Under the Tang rule, China became a cosmopolitan society by incorporating foreign cultural elements introduced via the Silk Road [83], like those found in the *Haiyao Bencao* (海藥本草, *Overseas Pharmacopoeia or Pharmacopoeia of Foreign Drugs*) compiled by Li Xun (855-930), a Chinese-born Persian physician [84]. The subsequent *Zheng Lei Ben Cao* (證類本草, *Materia Medica Arranged According to Pattern*) was compiled in 1108 AD during the Song Dynasty (960–1279) and included 1746 medicinals, among which are two marine macroalgae: *Gloiopeltis furcata* (Rhodophyta) and *Laminaria* (*Saccharina*) *japonica* (or sweet kelp, Phaeophyceae) [85,86]. The *Bencao Tujing* (圖經本草, *Illustrated Classic of Materia Medica*, 1061) and the *Zhu Fan Zhi* (诸蕃志, *A Description of Barbarian* Nations, 1225), also published under the Song rule, reflected the impact of the Maritime Silk Road seafaring on the evolution of Chinese medical knowledge [87,88,89]. Under the Mongol-led Yuan Dynasty (1271 to 1368), scientific ideas circulated inside the Empire via the Indian Ocean trade routes [90], and, as a result, the Islamic formulary *Huihui Yaofang* (回回藥方考釋, *Muslim Medicinal Recipes*) became a reference Imperial medical text [91,92]. During the Great Ming Dynasty (1368–1644), the court physician Li Shizhen (1518–1593) compiled the *Bencao Gangmu* (本草綱目, *Compendium of Materia Medica*), an outstanding piece of scientific literature describing 1892 medicinals. This text includes a detailed description of marine fauna and flora and stood until the nineteenth century as a reference for taxonomical classification in East Asia [93]. Notably, the *Bencao Gangmu* contains medical prescriptions based on seaweeds of the Ulvophyceae, Phaeophyceae, Florideohyceae, Trebouxiophyceae, and Bangiophyceae classes; cyanobacteria of the *Nostoc* genus; and seahorses [94,95].

The species *Hippocampus kuda* is known to produce an antitumor peptide with inhibitory activity on major intracellular signalling cascades: the nuclear factor kB (NF-kB)-mediated pathway, the Janus kinase 2/Signal Transducers and Activators of Transcription 3 (JAK2/STAT3) pathways, and the Jun N-terminal kinase (JNK)/p38 mitogen-activated Protein Kinase (p38 MAPK) pathway [96,97,98,99,100,101,102]. Seahorses are heavily used in TM, with an estimated annual consumption of approximately 250 tons in China and Hong Kong [103]. Such overexploitation, however, is posing a severe extinction threat to several *Hippocampus* species in both East Asia and Latin America [104], which are now included in the Appendix II of the Convention on International Trade in Endangered Species of Wild Fauna and Flora (CITES) [105] and in the Red List of the International Union for Conservation of Nature (IUCN) [106].

During the last imperial Dynasty of the Qing (1644–1911), European missionaries visited and established themselves in China, introducing the Christian faith and other Western cultural elements, including cartography. At the court of the Qing Emperor Shengzu, the Flemish Jesuit and astronomer Ferdinand Verbiest (1623–1688) published the *Kunyu Quantu* (坤舆全图, *Full Map of the World*) in 1674, one of several Chinese world maps produced in that era. Geography offered a glimpse of the outer world, attracting the attention of the traditionally self-centered and self-isolated Chinese civilization towards Western science (Figure 1) [107]. The Qing era was characterized by profound and irreversible changes in the Chinese society caused by violent political upheaval and by Western colonization, forcing the opening to foreign trade [108]. Furthermore, the end of the Imperial era was followed by the adoption of a universal “modern” science, although TM will never be fully replaced [109].

The opening of the Marine Biological Station of Amoy University in the 1930s under the guidance of foreign-trained Chinese biologists [110] marked a major leap forward in Chinese exploration of national marine biodiversity. These endeavours continue to the present day, with extensive bioprospecting activity being conducted in the South China Sea, leading to newly discovered marine lead compounds [111].

Much credit for the development of Chinese mariculture in the second half of the twentieth century goes to the US-trained marine botanist Cheng Kui Tseng (1909–2005), who worked at the Institute of Oceanology at Qingdao (Shandong province). This unsung hero of the 1940s modernization movement known as “Saving the Country by Means of Science” (科学救国) [112] contributed to the breaking of records of seaweed productivity achieved in the northern coastal Shandong Province [113].

Standing out from the crowd, China boasts a Modern Marine Materia Medica, a scientific encyclopaedia of marine medicinal organisms and of chemicals curated by the Key Laboratory of Marine Drugs of the Ministry of Education at the Ocean University of China in Qingdao. Besides holding tremendous scientific value, this project, supported by the special program “Project 908” (Comprehensive investigation and evaluation on the offshore oceans of China, *Zhong guo jin hai hai yang zong he diao cha yu ping jia*, 中国近海海洋综合调查与评价) [114], offers a privileged insight into one of the oldest surviving human civilizations, reflecting the uninterrupted continuity of Chinese culture throughout the millennia [115].

### The Yin Yang Dialectic of Autoimmunity

With the introduction of Western medicine into China in the middle seventeenth century, traditional Chinese medicine (TCM) started to evolve in the constant struggle between traditionalism and modernization [116]. In TCM, spiritual and scientific concepts coexist in a holistic discipline [117], which is in stark contrast with the reductionist Western approach. However, some TCM principles reflect key features of the immune system: balance, defense, holism, and circadian rhythms [118]. According to TCM, healthy immune functions require a harmonious equilibrium of *Yin* reserves (阴, organs, tissues, cells, and body fluids) and *Yang* (阳, physiological functions). The two are opposing forces constantly trying to win over one another. *Yin* and *Yang* are kept in a constant dynamic balance to preserve the *Qi* (气), the body’s vital energy [119], a concept analogous to *Pneuma* (breath of life, spirit) from classical antiquity [120]. A persistent *Yin* deficiency causes *Shanghuo* (上火), or heat syndrome, eventually creating an “excess of pathogenesis caused by deficiency” [121]. In a healthy individual, apoptosis regulates tissue homeostasis, maintaining *Yin* and *Yang* balance. When the capacity to remove apoptotic cells is overwhelmed by tissue degradation, the excessive exposure of auto-antigens to the immune system breaks immunological tolerance, triggering autoimmunity. Remarkably, several immune components embody the *Yin Yang* dialectic [122]. For instance, CD4+ T helper cells undergo dynamic functional specialization via cytokine-mediated signaling feedback [123]. Initially, a Th1 differentiation program is activated by interleukin (IL)-12 and interferon gamma (IFN-γ). Th1 cells later switch to producing the Th2-program cytokine IL-4 to prevent unrestrained Th1 proliferation.

Similarly, dendritic cells undergo divergent differentiation programs in response to chemical stimuli [124]. Instead, regulatory T cells—a lymphocyte population which suppresses immune responses and maintains self-tolerance—can convert to inflammatory Th17 cells [125]. Finally, individual cytokines can be either *Yin* or *Yang* elements, i.e., IL-6 is endowed with both pro- and anti-inflammatory functions depending on the context [126]. Strikingly, many TCM practices either stimulate or suppress immune functions [127,128,129]; therefore, this ancient medical art predates the discovery of the immune system and its complex functioning.

## 5. The Birth of Marine Microbiology: Sailing on the Ocean of Chemodiversity

The invention of optical instruments in the eighteenth century by Dutch lensmaker Antonie van Leeuwenhoeck and English polymath Robert Hooke enabled the discovery of marine microbes, marking a fundamental breakthrough in the appreciation of ocean biodiversity [130,131]. The ensuing scientific excitement stimulated the publication of *Die Bakterien des Meeres* (*Bacteria of the Sea*) in 1894 by German biologist Bernhard Fischer [132] and, later, of the treatise *Marine Microbiology: A Monograph on Hydrobacteriology* by American microbiologist Claude ZoBell [133]. Decades later, the discovery of marine cyanobacteria by John Waterbury and Sallie Chisholm [134,135] provided new insights into ocean primary productivity and revolutionized marine ecophysiology.

In the wake of the genomic and post-genomic eras, the exploration of the ocean by means of molecular techniques has revealed the staggering complexity of microbial biodiversity [136]. Metagenomics was first introduced in marine sciences by The Sorcerer II Global Ocean Sampling expedition led by American biotechnologist Craig Venter (2004–2006) [137,138,139]. The subsequent Malaspina (2010) and Tara Oceans expeditions (2009–2013) further revealed the richness of microbial life from the deep-sea environments [140,141,142]. The picture of the global ocean microbial genome was recently assembled in the KAUST Metagenome Analysis Platform (KMAP) Global Ocean Gene Catalog 1.0. This database contains 308.6 million gene clusters assembled from 2.102 metagenomes and represents an invaluable resource for the functional discovery of microbial metabolic pathways [143].

### 5.1. Prokaryotes and Metazoan-Associated Microbiota

Marine bacteria produce an arsenal of secondary metabolites used for inter- and intra-specific communication [144]. During evolution, key genes encoding polyketide synthases and non-ribosomal peptide synthetases have undergone extensive reshuffling within operons, creating a remarkable diversification of metabolic pathways [145]. A recent analysis of marine prokaryotic genomes revealed an astonishing complexity of biosynthetic gene clusters, although only a tiny fraction of secondary metabolites has been studied [146]. A major obstacle to microbial drug discovery, however, lies in the recalcitrance to culturability of marine isolates, since most species naturally grow in consortia [147,148]. Moreover, many secondary metabolites originally believed to be produced by marine invertebrates derive from the associated microbiota [149]. Mutualistic relationships between fungi and bacteria and metazoans (holobionts) are found in corals and sponges [150]. Strikingly, nearly all bioactive polyketides and peptides isolated from the *Theonella swinhoei* (porifera) holobiont are produced by the filamentous bacterial symbiont *Entotheonella* spp. [151,152,153]. Similarly, the anticancer molecule trabectedin was isolated from *Candidatus Endoecteinascidia frumentensis*, the bacterial symbiont of the sea squirt *Ecteinascidia turbinata* [154]. However, the study of secondary metabolites from mutualistic relationships is complicated by the strict host dependency of endosymbionts and the alteration of holobiont composition upon ex situ cultivation [155].

### 5.2. Fungi and Protists

Several *Talaromyces* fungal symbionts of algae and sponges [156] produce polyketides, alkaloids, terpenoids, peptides, and lipids, with reported anti-inflammatory potential [157,158]. Saprotrophic protists are emerging biofactories of immunomodulatory lipids, including the omega-3 polyunsaturated docosahexaenoic (DHA, C22:6 ω-3) and eicosapentaenoic acids (EPA, C20:5 ω-3) produced by the Thraustochytrid *Schizochytrium* sp. (reviewed in [159,160]), which is amenable to fermentation at a large scale in seawater and wastewater [161]. Notably, the anti-inflammatory properties of DHA compounds were recently reported in a clinical trial involving patients with rheumatoid arthritis [162].

### 5.3. Marine Eukaryotic Microalgae and Cyanoprokaryotes

Eukaryotic phytoplankton (hereafter microalgae) is potentially the richest resource for drug discovery and a promising platform for large-scale and low-cost production of high-value metabolites [163]. Microalgae comprise a vast group of photosynthetic microbes producing a huge repertoire of anti-inflammatory and immunomodulatory pigments and lipids [159,164,165,166,167].

Carotenoids are lipophilic pigments [168,169,170,171] designated as carotenes (lycopene and α- and β-carotene), which contribute to light-harvesting, and the oxygenated derivatives xanthophylls (or ketocarotenoids: astaxanthin, fucoxanthin and lutein), which are mainly involved in the detoxification of reactive oxidative species (ROS) generated by photosynthetic reactions. The ketocarotenoid astaxanthin is the microalgal pigment of greatest pharmacological value, being endowed with strong antioxidant capacity (extensively reviewed in [159]). The biological production of astaxanthin, however, is restrained by the slow growth of its native producer, the freshwater Chlorophyte *Haematococcus lacustris* (previously named *Haematococcus pluvialis*) [172]. Accordingly, the establishment of optimal cultivation strategies of *H. lacustris* and the domestication of high-yielding strains are key to accruing astaxanthin accumulation [173,174,175,176].

In response to stress, several microalgae synthesize DHA, EPA [177], while the chlorophyte *Tetraselmis chui* accumulates monogalactosyldiacylglycerols, which inhibit nitric oxide production [178,179,180,181]. The haptophyte *Tisochrysis lutea* (formerly known as *Isochrysis affinis galbana*) is the main DHA producer, although the eustigmatophytes *Microchloropsis salina*, *Nannochloropsis oceanica,* and *Microchloropsis gaditana* are emerging EPA producers, and species of the Pavlovophyceae family are sources of both carotenoids and functional lipids [182,183,184]. Diatoms are widely distributed eukaryotic phytoplankton [185] which produce valuable immunomodulatory pigments and lipids [186], particularly the genus *Thalassiosira*, which accumulates human-like prostaglandins in response to environmental stress [187,188,189,190].

At present, China is the major world producer of microalgal biomass for human consumption [191], while the US and several European countries are at the forefront of microalgal biotechnology research [192,193]. Bioprospecting for industrially relevant microalgae, instead, is conducted worldwide, mainly in inhospitable habitats, since extremophiles hyper-accumulate bioactive compounds [194] and display robust growth phenotypes under co-cultivation [195,196,197].

The full biotechnological exploitation of non-conventional microalgae, however, requires improvements in biomass yield and downstream processing to retrieve target metabolites [198,199].

Finally, marine cyanoprokaryotes are emerging sources of bioactive metabolites [200], several endowed with anti-inflammatory and immunomodulatory properties, including polysaccharides, phenols, flavonoids, and phycobiliproteins [201,202,203,204]. Notably, the light-harvesting pigment C-phycocyanin is a selective inhibitor of the enzyme cyclooxigenase-2 producing the pro-inflammatory mediator prostaglandin E_2_ [205,206,207,208].

## 6. Mining the Seabed for Novel Bioactive Compounds

Considered an azoic zone until the late nineteenth century [209], the deep sea has always stimulated the curiosity of scientists interested in the search for life in this mysterious environment. Starting with the British-led oceanographic dredging cruises of the H.M. SS. Porcupine and Lightning (1868–1870) [210,211] and the ensuing Challenger Atlantic voyage led by marine zoologist Sir Charles Wyville Thomson (1872–1876) [212], the abysses revealed remarkable biodiversity. These were followed by the French Travailleur and Talisman expeditions (1880–1883), during which barophilic microbes were collected and cultivated by Adolph-Adrien Certes, a disciple of Louis Pasteur [213,214].

The excitement for these newly discovered ecosystems was felt internationally, prompting the establishment of zoological stations dedicated to the study of marine biology and ecology [215]. Among these, the Stazione Zoologica of Naples (Italy), founded in 1872 by German zoologist Anton Dohrn, rapidly acquired international prestige and, today, is a leading institution in the field of marine bioprospecting and drug discovery research [216]. Between 1950 and 1952, the Danish-led Galathea Deep Sea Expedition (1950–1952) reached sampling depths of 10,000 m below sea level, revealing the existence of extremely barophilic bacteria [217,218] and prompting their cultivation in the laboratory [219]. In 1979, the manned submersible vessel *Alvin* enabled the discovery of hydrothermal vents in the Galápagos Rift of the East Pacific Ocean and of their associated communities of extremophile microbes [220].

Arguably, the deep sea is the last uncharted frontier for bioprospecting [221,222]. This extreme environment is home to thermophile, halophile, alkalophile, psychrophile, piezophile, and polyextremophile microorganisms, which produce a panoply of secondary metabolites (the chemical structures of recently identified lead compounds from deep-sea organisms are shown in Figure 2, while in Table 1, their biological activity and half-maximal inhibitory concentration, IC_50_, values are provided) [223,224,225,226]. Deep-sea biodiversity and chemodiversity are currently being investigated using approaches combining imaging, sampling, and genomics analysis with the creation of dedicated repositories, such as the MArine Bioprospecting PATent (MABPAT) Database, which provides free access to this constantly expanding biological landscape [227,228,229].

### 6.1. Deep-Sea Prokaryotes

Despite their physiological adaptation to extreme environmental conditions, deep-sea microbes can be easily cultured under laboratory conditions, enabling detailed investigation of their secondary metabolites and, recently, genetic engineering [230,231]. Bioprospecting for anti-inflammatory compounds of deep-sea bacteria resulted in the identification of a macrolactin derivative (7,13-epoxyl-macrolactin A, a 24-membered ring lactone, Figure 2.1) from *Bacillus subtilis* B5, which strongly inhibited pro-inflammatory gene expression and prevented the production of the inflammatory mediators interleukin-1β and IL-6 [232]. Another example of a bacterium-derived compound with immunomodulatory properties is the exopolysaccharide from *Planococcus rifietoensis* (described in Section 8.2) [233]. As in the case of shallow-water species of Porifera, symbioses between microbes and invertebrates are equally common in the deep sea, although less heterogeneous, as revealed by recent investigations into the sponge-associated microbiome [234]. Therefore, deep-sea holobionts are potential new sources of bioactive compounds awaiting characterization.

### 6.2. Deep-Sea Fungi

The advancement of deep-sea bioprospecting is reflected by the increasing number of newly identified bioactive compounds from fungal species [235], with the *Microbacterium*, *Dermacoccus, Streptomyces*, and *Verrucosispora* of the phylum actinomycota being the most studied genera [236]. Several ascomycota species also produce anti-inflammatory compounds with inhibitory activity against nitric oxide release. These include cyclopenol (a 7-membered 2,5-dioxopiperazine alkaloid, Figure 2.2), derived from *Aspergillus* sp.; the fusaric acid derivatives hepialiamides (Figure 2.3); and one novel hybrid polyketide hepialide (Figure 2.4) from *Samsoniella hepiali*. The latter also produces uridine, ergosterol, walterolactone A, (4R, 5S)-5-hydroxyhexan-4-olide, and myrothecol (Figure 2.5–10). Furthermore, *Acremonium* sp. and *Eutypella* sp. accumulate eremophilane-like sesquiterpenoids (Figure 2.11) [237,238,239,240,241], while the Ascomycetes *Penicillium oxalicum* and *chrysogenum* produce alkaloids and chrysamides (Figure 2.12–15), respectively, capable of suppressing the synthesis of pro-inflammatory mediators, including the potent cytokine IL-17 in vitro in the case of *P. chrysogenum* [242,243]. Finally, the Basidiomycete *Cystobasidium laryngis* has been shown to produce diphenazine derivatives with anti-neuroinflammatory properties (Figure 2.16) [244].

**Table 1 marinedrugs-22-00304-t001:** Recently discovered anti-inflammatory and immunomodulatory compounds from deep-sea microorganisms.

Molecule	Source Organism(s)	Biological Activity—Half Maximal Inhibitory Concentration (IC_50_)	Development Stage	Ref.
7,13-epoxyl-macrolactin A	*Bacillus subtilis* B5 (Gram-positive bacterium)	Suppression of *inducible nitric oxide synthase*, *IL-1β*, and *IL-6* expression in cultured activated murine macrophages (IC_50_ N.D.).	Preclinical trial	[232]
Extracellular Exopolysaccharide	*Planococcus rifietoensis* AP-5 (Gram-positive bacterium)	Stimulation of IL-10, IL-6, IL-1β, and TNF-α production by human cultured monocytes (IC_50_ N.D.).	Preclinical trial	[233]
Cyclopenol (7-membered 2,5-dioxopiperazine alkaloid)	*Aspergillus* sp. (Ascomycota)	Suppression of nitric oxide release by cultured activated murine macrophages via inhibition of the NF-κB pathway. Down-regulation of *inducible nitric oxide synthase*, *IL-1β* and *IL-6* in cultured activated murine microglia (IC_50_ 30 µM).	Preclinical trial	[237]
Hepialiamides (fusaric acid derivatives)	*Samsoniella hepiali* W7 (Ascomycota)	Suppression of nitric oxide release by cultured activated murine microglia (IC_50_ 1 µM).	Preclinical trial	[238]
Polyketide hepialide	*Samsoniella hepiali* W7 (Ascomycota)	Suppression of nitric oxide release by cultured activated murine microglia (IC_50_ 1 µM).	Preclinical trial	[238]
5′-O-acetyladenosine, uridine, ergosterol, walterolactone A	*Samsoniella hepiali* W7 (Ascomycota)	Suppression of nitric oxide release by cultured activated murine microglia (IC_50_ 1 µM).	Preclinical trial	[238]
(4R,5S)-5-hydroxyhexan-4-olide	*Samsoniella hepiali* W7 (Ascomycota)	Suppression of nitric oxide release by cultured activated murine microglia (IC_50_ 426 nM).	Preclinical trial	[238]
2-benzoyl tetrahydrofuran enantiomers (−)-1S-myrothecol, (+)-1R-myrothecol	*Myrothecium* sp. (Ascomycota)	Suppression of nitric oxide release by cultured activated murine macrophages (IC_50_ 1.20 and 1.41 µgmL^−1^).	Preclinical trial	[239]
Acremeremophilanes Eremophilane-Type Sesquiterpenoids	*Acremonium* sp. (Ascomycota)	Suppression of nitric oxide release by cultured activated murine macrophages (IC_50_ 8 to 45 μM).	Preclinical trial	[240]
Eremophilane-Type Sesquiterpenoids	*Eutypella* sp. (Ascomycota)	Suppression of nitric oxide production by cultured activated murine macrophages (IC_50_ 8 to >50 μM).	Preclinical trial	[241]
Oxaline (A), isorhodoptilometrin (B), and 5-hydroxy-7-(2′-hydroxypropyl)-2-methyl-chromone (C).	*Penicillium oxalicum* (Ascomycota)	Suppression of nitric oxide and prostaglandin E_2_ production by cultured murine microglia cells. Down-regulation of *inducible nitric oxide synthase* and *cyclo-oxygenase-2* expression. Inhibition of TNF-α, IL-1β, IL-6, and IL-12 production via interference with the NF-κB and MAPK pathways (IC_50_ A 9, B 15, and C 75 μM).	Preclinical trial	[242]
Dimeric nitrophenyl trans-epoxyamides Chrysamides A-C	*Penicillium chrysogenum* (Ascomycota)	Suppression pro-inflammatory cytokine IL-17 production by cultured murine naïve T cells (IC_50_ C 75 μM).	Preclinical trial	[243]
Phenazostatins (Diphenazine derivatives)	*Cystobasidium laryngis* (Basidiomycota)	Suppression of nitric oxide and IL-6 production by activated murine macrophages in vitro via inhibition of NF-κB pathway. Suppression of *IL-1β*, *IL-6,* and *inducible nitric oxide synthase* expression in cultured murine microglia cells (IC_50_ 0,30–170 μM).	Preclinical trial	[244]

**Figure 2 marinedrugs-22-00304-f002:**
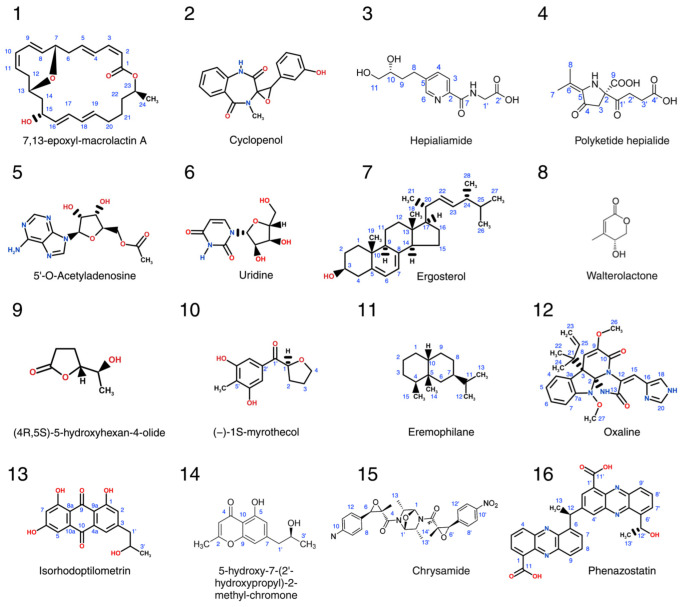
Chemical structures of recently identified immunomodulatory and anti-inflammatory compounds from deep-sea organisms described in Table 1. (**1**) 7,13-epoxyl-macrolactin A from *Bacillus subtilis* B5; (**2**) cyclopenol from *Aspergillus* sp. [237]; (**3**–**9**) hepialiamide, polyketide hepialide, 5′-O-Acetyladenosine, uridine, ergosterol, walterolactone A, and (4R, 5S)-5-hydroxyhexan-4-olide from *Samsoniella hepiali* W7; (**10**) myrothecol from *Myrothecium* sp. [239]; (**11**) eremophilane from *Acremonium* sp. and *Eutypella* sp. [240,241] (**12**–**14**) oxaline, isorhodoptilometrin, and 5-hydroxy-7-(2′-hydroxypropyl)-2-methyl-chromone from *Penicillium oxalicum* [242]; (**15**) chrysamide from *Penicillium chrysogenum* [243]; and (**16**) phenazostatin from *Cystobasidium laryngis* [244].

## 7. Emerging Marine Immunomodulatory Lead Compounds

### 7.1. Seaweeds (Macroalgae)

Seaweeds, or macroalgae, are plant-like multicellular phototrophs found in intertidal waters attached to rocks or floating free in the open sea [194]. The edible red seaweed *Porphyra dentata* (Rhodophyta, Bangiales) is widely used worldwide in TM and contains phenolic compounds which inhibit the synthesis of the pro-inflammatory signaling molecule nitric oxide (NO) and suppress the NF-kB pathway in vitro [245].

Several species are farmed in the ocean due to their highly nutritious composition and content of bioactive compounds, mainly sulfated polysaccharides [246,247,248]. For instance, porphyran, a sulfated galactan from red seaweeds (Porphyra), stimulates the immune function and apoptotic/autophagic processes, and is considered a candidate anti-cancer drug (IC_50_ 20 μg/mL) [249]. Similarly, the polysaccharide fraction from *Lithothamnion muelleri* (Hapalidiaceae) displayed immunomodulatory activity by inhibiting the synthesis of pro-inflammatory chemokines in an animal model of arthritis [250]. Moreover, fucoidan and ulvan extracted from the brown macroalga (kelp) *Undaria pinnatifida* and the green alga *Ulva lactuca* (IC_50_ 623.58–785.48 µg/mL), respectively, are potent immunostimulators and antioxidants, and thus, have potential applications to reduce the side effects of immunosuppressive therapies [251,252]. Anti-inflammatory effects were reported for polysaccharide extracts from *Halimeda tuna* (Ulvophyceae) [10] and *Posidonia oceanica* (*Alismatales*) [253], and for carrageenans derived from different red seaweeds (*Chondrus crispus*, *Ahnfeltiopsis devoniensis*, *Sarcodiotheca gaudichaudii*, and *Palmaria palmata*) [254]. Finally, polysaccharide fucosterol and phlorotannins from the brown macroalgae *Sargassum wightii* and *Eisenia bicyclis* (eckol, dieckol and 7-phloroeckol; IC_50_ 52.86, 51.42 and 26.87 μg/mL, respectively) suppressed the release of pro-inflammatory mediators in animal models of arthritis [255,256].

### 7.2. Invertebrates

Several marine invertebrate phyla are known producers of anti-inflammatory and immunomodulatory compounds [257,258]. Corals are colonial organisms of the class Anthozoa (Cnidaria), typically found in reef ecosystems in tropical and sub-tropical water. Coral bioprospecting is mainly focussed on the octocorallia order *Alcyonacea* (Gorgonians or soft corals), which is known to produce a vast repertoire of secondary metabolites of pharmacological relevance [259], mainly immunomodulatory lipid derivatives and terpenoids [260,261]. For instance, the sesquiterpenes (C_15_H_24_) capnellenes, isolated from the species *Capnella imbricata,* inhibited the expression of the pro-inflammatory enzymes inducible nitric oxide synthase and cyclooxygenase-2 in cultured macrophages (used at 10 μM) [262,263]. The Caribbean gorgonian *Plexaura homomalla*, instead, emerged as the highest natural producer of mammalian-like prostaglandins, which are hormone-like oxygenated metabolites of C20 fatty acids involved in the modulation and resolution of inflammation [264,265]. Similarly, diterpenes (C_20_H_32_) isolated from the Formosan gorgonians *Briareum excavatum* (excavatolide, or excavatoid B used at 50 μM, E and F with ED_50_ > 40 μg/mL) and *Sinularia querciformis* (11-*epi*-sinulariolide acetate; IC_50_ 50 μM) inhibited the synthesis of several pro-inflammatory mediators in arthritis animal models [266,267,268]. Lastly, peptides isolated from the venom of the jellyfish *Pelagia noctiluca* (Pelagiidae) strongly suppressed nitric oxide production in vitro (used at 50 μg/mL) [269].

Ascidians (Chordata, subphylum Tunicata) have emerged as novel sources of bioactive compounds, mainly derived from their innate immune system [270]. Recently, a synthetic peptide derived from the sea squirt *Styela clava* was shown to exert both antimicrobial and immunomodulatory activities in animal models. In particular, the clavanin-MO peptide (used at 2 μM) promoted the synthesis of the anti-inflammatory cytokine IL-10 while suppressing the release of the pro-inflammatory factors IL-12 and tumor necrosis factor-α (TNF-α) upon bacterial infection [271]. Another study described the in vitro inhibitory effects of chemical inhibitors based on the structure of metabolites from *Herdmania momus* against multiple pro-inflammatory enzymes and the release of pro-inflammatory cytokines in activated macrophages (IC_50_ between 7.59 and 39.20 μM) [272]. Finally, although the chemical nature of the bioactive compound(s) still awaits characterization, extracts of the Indonesian ascidian *Polycarpa aurata* acted as hydrogen sulfide donors in vitro, suppressing the pro-inflammatory response of cultured macrophages (used at 50 µg/mL) [273].

Gastropods of the family Muricidae (Mollusca) are known to produce mucus-containing bioactive molecules. A recent study showed the immunomodulatory effects of the mucus of *Bolinus brandaris*, suggesting that a still-uncharacterized compound could trigger the immune system against cancerous cells by inducing monocyte differentiation (IC_50_ ranging between ≤1 and ~10 µg/mL) [274]. Moreover, lipids extracted from the mussel *Mytilus coruscus* exerted a strong anti-inflammatory effect in a murine arthritis model, reducing the levels of the pro-inflammatory mediators leukotriene B₄, prostaglandin E₂, and thromboxane B₂, but also of the cytokines IL-1β, IL-6, interferon-γ, and tumor necrosis factor-α [275]. Notably, a similar preparation was tested in a clinical trial involving rheumatoid arthritis patients with similar outcomes [276]. Finally, the two sea hares *Aplysia fasciata* and *Aplysia punctata* (Anaspidea) were shown to produce immunomodulatory lipids which suppressed the activity of pro-inflammatory enzymes and nitric oxide production in vitro (IC_50_ 77 and 74 µg/mL, respectively) [277].

Echinodermata are another phylum of marine animals which synthesize bioactive compounds with immunomodulatory properties [278]. Echinozoa, or sea urchins, are the best-studied group, with a recent example of an anti-inflammatory lead compound identified in *Scaphechinus mirabilis* (described in detail in Section 8.1) [279]. Another example includes *Isostichopus badionotus*, whose extracts suppress the expression of pro-inflammatory genes in vivo [280]. Moreover, a recent study described the anti-inflammatory activity of the protein cargo of extracellular vesicles of the sea cucumber (Holothuroidea) *Stichopus japonicus*, showing a strong inhibition of the release of pro-inflammatory cytokines by cultured synoviocytes, a cell type involved in the pathogenesis of osteoarthritis (used at 10 µg/mL) [281].

Lastly, the phylum Porifera contains several species of marine sponges, especially of the genus *Hyrtios*, which are sources of bioactive compounds, mainly alkaloids and terpenoids [282]. Early studies have reported the in vitro and in vivo anti-inflammatory activity of scalaristerol (5alpha,8alpha-dihydroxycholest-6-en-3beta-ol) and callysterol (ergosta-5,11-dien-3beta-ol), isolated from *Scalarispongia aqabaensis* (Thorectidae) and *Callyspongia siphonella* (Callyspongidae), respectively [283], and from *Aplysina caissara* (Aplysinidae), *Haliclona* sp. (Chaliunidae), and *Dragmacidon reticulatum* (Axinellidae), although the chemical nature of their bioactive compounds could not be identified [284]. Recently, fistularins (bromotyrosine acids) isolated from *Ecionemia acervus* (Ancorinidae) strongly inhibited the activity of pro-inflammatory enzymes and the release of inflammatory cytokines from cultured macrophages (tested range between 5 and μM) [285]. Similarly, the brominated alkaloid aeroplysinin derived from *Aplysina aerophoba* displayed inhibitory effects on cultured vascular endothelial cells by suppressing the NF-kB pathway (IC_50_ 3 µM) [286]. Furthermore, the norditerpene dihydrogracilin A, derived from the Antarctic sponge *Dendrilla membranosa* (Darwinellidae), suppressed the NF-kB pathway in cultured human peripheral blood mononuclear cells and dampened the production of the pro-inflammatory cytokine IL-6 (tested range between 0.3 and 10 µM) [287]. Finally, lipids extracted from *Halichondria sitiens* exerted immunomodulatory effects on cultured dendritic cells (used at 10 µg/mL) by suppressing the secretion on the pro-inflammatory cytokines IL-12 and Il-6, but also prevented the production of IFN-γ by CD4+ T lymphocytes, thus blocking the so-called Th1-type immune response (explained in detail in Section 8.2) [288].

### 7.3. Mangrove Habitats

Mangrove forests are threatened coastal habitats at the interface between terrestrial and marine tropical environments in which salt-tolerant plants (halophytes) create unique ecosystems hosting a plethora of interacting microorganisms [289]. Mangroves are widely consumed in the TM of Southern India [290,291,292,293], and recent ethnopharmacological studies have isolated several phytochemicals with anti-inflammatory and immunomodulatory properties from *Aegiceras corniculatum* [294], *Rhizophora mucronata* [295], and *Sonneratia apetala* [296]. Notably, the leaf extracts from *A. corniculatum* inhibited the production of pro-inflammatory cytokines (TNF-α, IL-6, and IL-12) by in vitro cultured immune cells [295], while the extracts from *Lumnitzera racemosa* displayed anti-angiogenic properties (IC_50_ ranging between 2,57 and 4,95 µM) [297]. Moreover, agalloide terpenoids from *Ceriops decandra* (tested at 100 μM), and, mainly, *Excoecaria agallocha*, suppressed NF-kB pathway activation [298,299,300]. Besides providing new phytopharmaceuticals, mangrove forests are suitable habitats for bioprospecting microbial compounds [301]. For instance, two new sesquiterpenoid derivatives (elgonenes M and N used at 5 and 20 μM, respectively) were identified in the fungus *Roussoella* sp. after isolation from a mangrove sediment, which inhibited the synthesis of pro-inflammatory cytokines by cultured immune cells [302].

## 8. Marine Pharmacology, *Quo Vadis?*

Several marine lead compounds are currently being assessed in pre-clinical and clinical studies, while seven marine drugs have already received “first-in-class” status, i.e., are endowed with “new and unique mechanism(s) of action” [303,304,305,306]. Most marine drugs in clinical use find application in cancer immunotherapy, as antibody–drug conjugates and in the management of chronic inflammatory conditions (reviewed in detail in [307,308]). Two examples of recently identified marine molecules are provided in the following paragraphs.

### 8.1. The Sea Urchin Echinochrome A and Its Applications in Systemic Sclerosis

Systemic sclerosis (SSc) is a rare immune-mediated connective tissue disease characterized by microvascular damage followed by aberrant autoimmune responses of the skin and internal organs, including the gastrointestinal tract, kidneys, lungs, and heart [309]. Fibrosis is a hallmark of SSc pathogenesis. This process is driven by activated pro-fibrotic myofibroblasts, highly differentiated cells which produce contractile proteins such as alpha-smooth muscle actin, resulting in excessive extracellular matrix deposition [310]. Although myofibroblasts contribute to the physiological process of wound healing in damaged tissues, their aberrant activation contributes to diffuse fibrosis and chronic inflammation [311,312]. Innate immune cells—particularly monocytes and macrophages—are established mediators of the fibrotic process in SSc [313]. Therefore, the discovery of novel immunomodulatory and anti-fibrotic molecules is of great clinical relevance for slowing SSc progression.

The marine compound Echinochrome A (6-ethyl-2,3,5,7,8-pentahydroxy-1,4-naphthoquinone, EchA) is a natural pigment from the echinoderm *Scaphechinus mirabilis* [314] endowed with antioxidant anti-fibrotic properties [315,316,317,318]. Recently, it was reported that EchA reduced collagen deposition and alleviated dermal thickness in an SSc animal model [279]. In this study, bleomycin was inoculated in the mouse skin for three weeks to induce dermal injury and to activate pro-inflammatory immune cells. The administration of EchA suppressed different mechanisms involved in the fibrotic process, including fibroblast activation and myofibroblast maturation; tumor growth factor (TGF)-β1-mediated expression of smooth muscle actin; phosphorylation of pro-fibrotic transcription factors in skin fibroblasts; and, finally, differentiation of macrophages into both M1 and M2 cells (Figure 3). These effects resulted in lower serum concentrations of pro-inflammatory cytokines TNF-α and IFN-γ. Overall, EchA is a promising marine lead compound with anti-fibrotic properties and, thus, potential application in the clinical management of SSc. Recently, a novel administration system based on polymeric nanofibers was developed to improve the water solubility of EchA. This pharmacological advance is expected to promote the controlled release of the drug, enhancing its bioavailability [319]. 

### 8.2. Deep-Sea Bacteria Exopolysaccharides and Their Applications in Cancer Immunotherapy

The interplay between immunity and tumorigenesis is a cornerstone of cancer biology, since the immune system exerts a multifaceted influence in terms of thwarting tumor initiation, progression, and metastasis [320]. Cancer immunotherapy emerged in the late twentieth century with the observation by American physician William Coley that sarcomas shrunk following inoculation of the tumor mass with killed bacteria [321]. It is now well established that tumor recognition and rejection by the immune system involve a complex dialogue between adaptive and innate immune cell types, including CD8+ cytotoxic T cells, CD4+ helper T (Th) cells (Th1, Th2, and Th17 lineages), regulatory T cells (Tregs), and myeloid-derived suppressor cells [322]. Moreover, macrophages—highly adaptable phagocytic immune cells [323]—can act as antigen-presenting cells (APCs) and differentiate into classically activated (M1) and alternatively activated (M2) types, with the former promoting inflammation and the latter fostering tissue repair [324]. M1 and M2, however, are the extremes of a broader cell type spectrum [325,326,327], since macrophages are known to interfere with tumorigenesis by influencing angiogenesis, fibrosis, and tumor cell phagocytosis. Moreover, macrophages orchestrate immunosurveillance by expressing costimulatory molecules like CD86 (B7-2) or T cell inhibitory molecules and promote the recruitment of immunosuppressive T-reg cells [328].

CD86 presented by APCs can bind either to cytotoxic T lymphocyte antigen 4 (CTLA-4) or CD28 on the surface of CD4+ and CD8+ T cells, causing their inhibition or activation, respectively [329]. Notably, upon cytokine signaling, APCs mediate the “immune synapse”, activating cytotoxic CD8+ T cells thanks the stimulatory and inhibitory receptors programmed cell death 1 (PD-1) and CTLA-4 [330]. Despite immune surveillance, neoplastic cells can still escape the immune system defense mechanisms [331]. Immunotherapy aims at overcoming this phenomenon by unleashing the host immune system against malignant cells. At present, immunotherapy approaches have been successful in promoting positive clinical responses across multiple cancer types [331].

The discovery of immune checkpoint inhibitors by American physician D. R. Leach prompted the use of antibodies to block CTLA-4 and trigger robust immune responses to achieve tumor shrinkage [332]. Currently, immune checkpoint inhibitors such as anti-CTLA-4, anti-PD-1, and anti-PD-L1 are regularly used in clinical practice to target regulatory pathways in T cells, essentially to reactivate the immune response against malignant cells [333,334]. However, the consequence of excessive activation of the immune system is the onset of autoimmune diseases such as rheumatic polymyalgia or serum-negative arthritis [335]. In this context, the dysregulation of the delicate balance between M1/M2 macrophages contributes to the pathogenesis of autoimmune diseases such as rheumatoid arthritis [336]. The understanding of the complex interplay between cancer and autoimmunity represents a continuously advancing area of study. Therefore, the identification of novel immunoactive molecules is crucial to developing new therapeutic strategies.

A recent study described the immunostimulating effect of a marine exopolysaccharide (EPS) produced by the deep-sea psychrotolerant Gram-positive bacterium *Planococcus rifietoensis* AP-5 [233]. This compound was tested on in vitro-cultured THP-1 monocytes differentiated into macrophage-like cells and treated with different EPS concentrations (5, 10, 20, 50, and 100 μg/mL). The authors reported negligible cell toxicity at low dosages, but increased phagocytic activity and high cytokine (IL-10, IL-6, IL-1β, and TNF-α) production, suggesting strong immunoregulatory properties of EPS on innate immune responses and, thus, a potential application of this marine polysaccharide as a complementary agent in cancer immunotherapy (Figure 4).

### 8.3. Synthetic Biology and Molecular Pharming in Microalgae

The genetic manipulation of microalgal genomes represents a booming field in biotechnology, projecting photosynthetic microbes as viable alternatives to conventional heterotrophic hosts (bacteria and yeasts) for the production of high-value recombinant therapeutics [337,338]. Advanced genetic tools are constantly being developed to: (i) achieve high-expression of foreign DNA sequences coupled to synthetic cis-acting regulatory elements [339,340]; (ii) introduce multigene expression constructs [341]; (iii) conduct iterative editing interventions in the nuclear genome [342]; and (iv) exploit the chloroplast genome for metabolic engineering and production of recombinant proteins [343,344]. Microalgae are suitable platforms for producing recombinant protein-based therapeutics since they perform eukaryotic post-translational modifications and can be engineered to secrete heterologous products in the cultivation media [345]. Different classes of recombinant therapeutics can be produced in microalgae, including full-length antibodies [346], anti-cancer cytokines [347], and immune receptors [348].

Furthermore, genetic egineering can be employed to enhance the yield of functional metabolites. For instance, endogenous metabolic circuits can be rewired to hyperaccumulate specific pathway intermediates or modified to accumulate non-native metabolites. Alternatively, entirely new biosynthetic pathways can be introduced to produce exotic metabolites starting from endogenous substrates [349,350]. Metabolic engineering in the model freshwater chlorophyte *Chlamydomonas reinhardtii* resulted in the synthesis of the non-native ketocarotenoid astaxanthin via overexpression of two heterologous biosynthetic genes [351] and via CRISPR-Cas9-based gene inactivation coupled to transgenesis [352]. In this respect, the recent genome annotation of *Haematococcus lacustris* is expected to facilitate the genetic engineering of astaxanthin accumulation, both in the native producer and in heterologous hosts [353].

Indeed, although most genetic tools have been established in *Chlamydomonas reinhardtii*, engineering strategies are currently being tested in non-conventional strains, including the marine rhodophyte *Porphyridium purpureum*, in which the expression of a glycosylated viral antigen was recently reported [354]. This is expected to significantly expand the range of therapeutic uses of microalgae, with respect to both recombinant protein expression [355] and hyper-accumulation of high-value pigments and lipids [356,357,358,359]. Finally, current developments of synthetic biology in marine cyanobacteria are expected to introduce significant novelties into the biomanufacturing of therapeutics in these highly productive photosynthetic microbes [360,361].

### 8.4. The Marine Viral Dark Matter and Its Potential for Medical Biotechnology

Despite not finding a place in the tree of life, viruses are the undisputed engines of evolution in the marine biosphere and are major drivers of its biogeochemical cycles [362,363]. The genetic complexity of the “marine viral dark matter” is just beginning to surface through recent studies [364], and is expected to bring about not only new insights into ecophysiological dynamics, but also innovations in biomedicine [365,366]. Historically, viruses have been a source of inspiration in the development of biomedical applications like vaccine production, cellular transfection, and phage therapy, to name a few. Of particular interest are phytoplankton-infecting viruses like Phycodnaviridae [367] and Cyanophages [368], infecting marine eukaryotic microalgae and cyanoprokaryotes, respectively. Moreover, certain structural features of viral proteins have broader relevance for biotechnology. One example is inteins, self-cleavable protein splicing elements enriched in marine viral genomes, which have been engineered into valuable biotechnological tools [369,370]. However, due to the scarcity of reports describing the use of marine viruses in biotechnology, at present, it is difficult to predict their impact on future developments of biomedical applications.

## 9. Conclusions

Marine pharmacology has come a long way, from superstitious practices to present-day high-throughput drug discovery pipelines. During the last three decades, the bioprospecting of marine environments has identified hundreds of lead compounds with potential applications in the clinical management of chronic inflammatory diseases and cancer. Several of these will likely be implemented as complementary agents in the clinical practice along with already established therapeutics, such as anti-cytokines, monoclonal antibodies, and immune checkpoint inhibitors.

It should be noted, however, that the developing “ocean blue economy” is threatening marine biodiversity due to the intense activities of shipping, transportation, fisheries, tourism, and renewable energy production. Among these, seabed mining has been proposed as a severe cause of biodiversity erosion [371,372,373,374], and the ecological impact of this industrial activity was revealed by metagenomics analysis showing a reduction of deep-sea microbial biodiversity [375]. Indeed, in contrast with land environments, the high seas are still a largely ungoverned and vast “no man’s land” lacking sustainable planning for resource management [376]. 

Moreover, anthropogenic climate change has manifold impacts on marine ecophysiology. It should be noted that the distribution of global marine plankton follows latitudinal gradients, with a steady decline towards the poles. On the one hand, ocean warming is expected to cause a tropicalization of plankton diversity in temperate and polar waters, putting at risk these under-explored fragile ecosystems and, thus, precluding future bioprospecting endeavours [377]. On the other hand, ocean acidification, a direct consequence of rising atmospheric CO_2_ levels, is suggested to interfere with the metabolism of marine flora, particularly of seaweeds, affecting their polysaccharide, fatty acid, and secondary metabolite composition and profile and potentially altering their reported pharmacological uses [378]. Therefore, the ultimate frontier of marine pharmacology lies in the development of sustainable biomanufacturing platforms of therapeutic compounds, away from low-yielding and threatened natural producers, through synthetic biology approaches in photosynthetic microbes.

## Figures and Tables

**Figure 1 marinedrugs-22-00304-f001:**
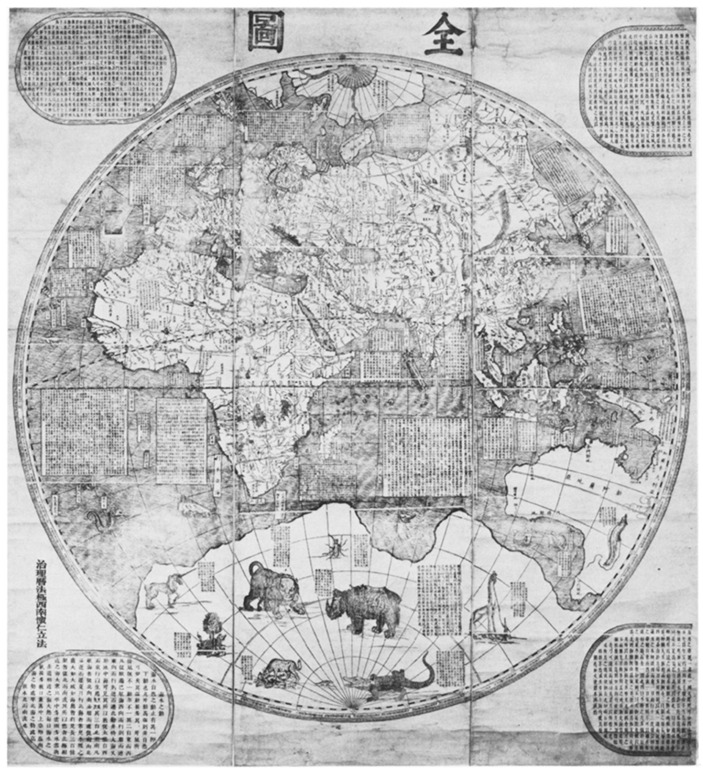
Marine pharmacology has come a long way, from superstitious practices to present-day high-throughput drug discovery pipelines. During the last three decades, the bioprospecting of marine environments has identified hundreds of lead compounds with potential applications in the clinical management of chronic inflammatory diseases and cancer. Historically, the East and West established their own corpuses of marine materia medica. During the last Chinese Imperial Dynasty of the Qing (1644–1911), European missionaries visited and established themselves in China, introducing the Christian faith and other Western cultural elements, including cartography. At the court of the Qing Emperor Shengzu, the Flemish Jesuit and astronomer Ferdinand Verbiest (1623–1688) published the Kunyu Quantu (坤舆全图, *Full Map of the World*) in 1674, one of several Chinese world maps produced in that era. Geography offered a glimpse into the outer world, attracting the attention of the traditionally self-centered and self-isolated Chinese civilization towards Western science. In the modern era, these two distant cultural worlds began a cross-fertilization of knowledge and, today, they together contribute to advancing the applications of marine resources in human health. Reproduced from [107].

**Figure 3 marinedrugs-22-00304-f003:**
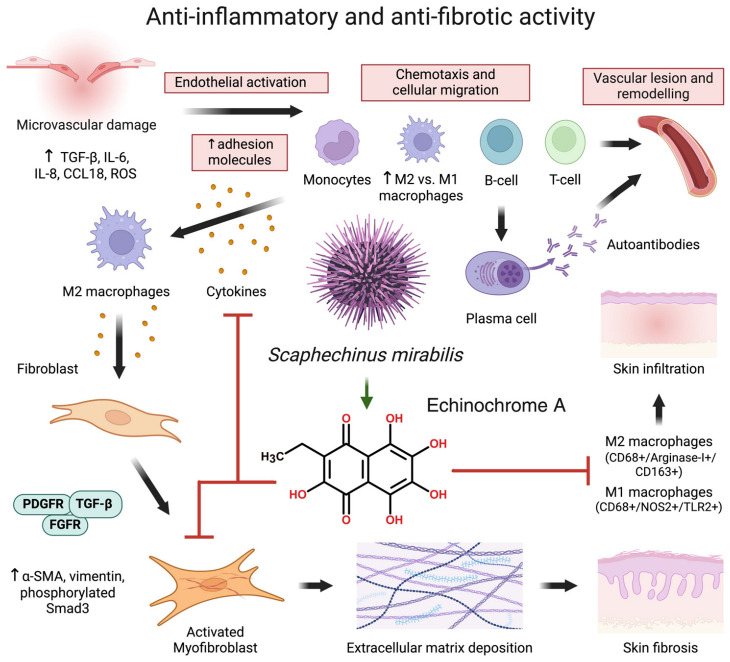
Antifibrotic activity of EchA in SSc model. EchA reduces skin cell infiltration (M1 and M2 macrophages) and myofibroblast activation, ameliorating skin thickness. Cytokines (IL), cluster of differentiation (CD), reactive oxygen species (ROS), protein-coupled receptor signaling pathway (CCL), transforming growth factor (TGF-β), platelet-derived growth factor receptors (PDGF-Rs), fibroblast growth factor receptor (FGFRs), and Echinochrome A (EchA). Figure created with Biorender (accessed on 12 June 2024), based on [279].

**Figure 4 marinedrugs-22-00304-f004:**
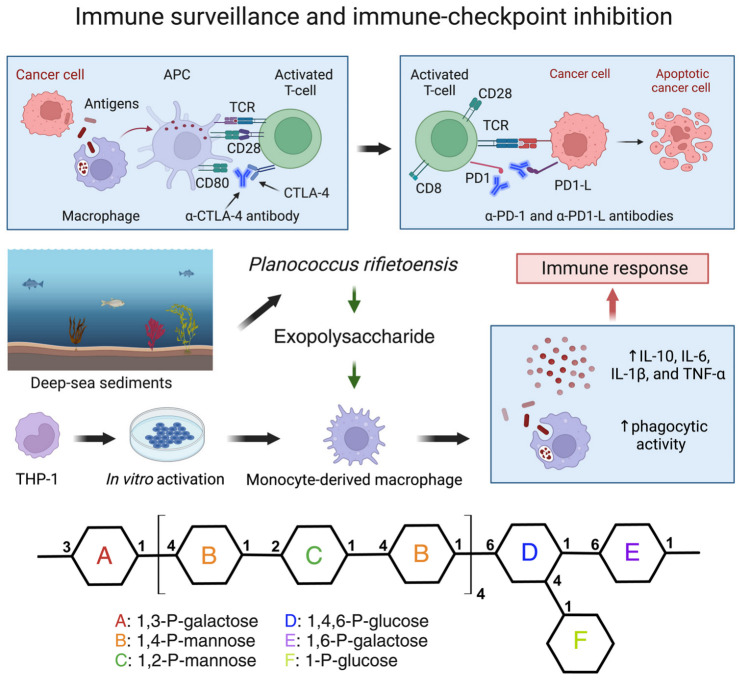
Immune surveillance and immunotherapy with immune checkpoint inhibitors. Illustration of in vitro EPS immunostimulant effect on monocyte-derived macrophages. Cytotoxic T-lymphocyte-associated protein 4 (CTLA-4), programmed cell death 1 (PD-1), antigen-presenting cell (APC), cytotoxic T-Lymphocyte antigen 4 (CTLA4), T cell receptor (TCR), cytokines (IL), cluster of differentiation (CD), human monocytic cell line (THP-1), tumor necrosis factor (TNF), phorbol 12-myristate13-acetate (PMA), and extracellular polysaccharides s(EPS). Figure created with Biorender (accessed on 12 June 2024), based on [233].

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
