# Peer review of "Immunomodulatory Compounds from the Sea: From the Origins to a Modern Marine Pharmacopoeia"

_marinedrugs, 2024, doi:10.3390/md22070304_

Round 1
Reviewer 1 Report
Comments and Suggestions for Authors
In this review, authors reviewed the historical development of marine pharmacology in the Eastern and Western worlds and describe recent discoveries of anti-inflammatory and immunomodulatory compounds. This is an interesting overview, the authors have summarised a large number of references and have put in an extraordinary amount of work. However, I think there are some concerns with this paper.
1. Even though this is a review, I still think there are too many keywords in this paper.
2. The format of this article is not the standard format for marine drugs, and the structure of this article looks a little strange.
3. There is a lot of unaligned text.
4. Line 686, Marine Pharmacology, Quo Vadis? This is an interesting topic, but please consider that many non-European and American readers, may not have a grounding in Latin, are also unfamiliar with Henryk Sienkiwicz.
5. Line 8.5, I think the conclusion deserves its section.
6. Acknowledgements should not normally be included in a separate section. The structure of this paper makes it a little strange to read, unlike a thesis.
7. Thanks to the authors for their hard work, but I think this article is too long. Especially chapters 2-5. I would suggest that chapters 2-5 be drastically reduced. Many of which are not helpful to this review.
8. With reference to the previous question, the corresponding author is asked to rearrange the structure and length of this paper.
9. I suggest that a table of recently discovered anti-inflammatory and immunomodulatory compounds from the deep sea would give the reader a better understanding of this paper.
10. There are also some detail errors in the article, line 604, Instead should be instead. In addition, the sections of this paper are not written uniformly and may have been done by different authors; the corresponding author is kindly requested to revise the entire text.
Author Response
In this review, authors reviewed the historical development of marine pharmacology in the Eastern and Western worlds and describe recent discoveries of anti-inflammatory and immunomodulatory compounds. This is an interesting overview, the authors have summarised a large number of references and have put in an extraordinary amount of work. However, I think there are some concerns with this paper.
EAC: Thank you for acknowledging the value of our manuscript. Please find below replies to your comments. We hope we have addressed all your concerns in the newly submitted version of the manuscript.
1 Even though this is a review, I still think there are too many keywords in this paper.
EAC: Thank you for this observation. We have now reduced the number of keywords to the following ten: bioprospecting; inflammation; autoimmunity; synthetic biology; drug discovery; genetic engineering; immunomodulation; deep sea; systemic sclerosis; rheumatoid arthritis
2 The format of this article is not the standard format for marine drugs, and the structure of this article looks a little strange.
3 There is a lot of unaligned text.
EAC: Thank you for this observation. We believe that the editorial team can fix this issue.
4 Line 686, Marine Pharmacology, Quo Vadis? This is an interesting topic, but please consider that many non-European and American readers, may not have a grounding in Latin, are also unfamiliar with Henryk Sienkiwicz.
EAC: Thank you for this observation. Although we understand your point, we would like to keep this Latin expression in the text.
5 Line 8.5, I think the conclusion deserves its section.
EAC: We have accordingly modified the section numbering and now the conclusions are section number 9.
6 Acknowledgements should not normally be included in a separate section. The structure of this paper makes it a little strange to read, unlike a thesis.
EAC: Done, thanks. This section was moved after the reference list.
7 Thanks to the authors for their hard work, but I think this article is too long. Especially chapters 2-5. I would suggest that chapters 2-5 be drastically reduced. Many of which are not helpful to this review.
8 With reference to the previous question, the corresponding author is asked to rearrange the structure and length of this paper.
EAC: Thank you for both suggestions. We have significantly shortened these 4 sections leaving only relevant information.
9 I suggest that a table of recently discovered anti-inflammatory and immunomodulatory compounds from the deep sea would give the reader a better understanding of this paper.
EAC: We have created a table summarizing this information based on your suggestion.
10 There are also some detail errors in the article, line 604, Instead should be instead. In addition, the sections of this paper are not written uniformly and may have been done by different authors; the corresponding author is kindly requested to revise the entire text.
EAC: Thank you for this observation. We have removed the word “instead” from the new manuscript version.
Reviewer 2 Report
Comments and Suggestions for Authors
The topic of this manuscript is quite interesting, and this review is important. The authors have profound knowledge on the origins of marine pharmacopoeia, which was revealed by the detailed introduction. However, as for the modern era, the review on the immunomodulatory marine compounds was lack of extensive examples and detail information. Of course, this manuscript is worth publishing in this journal.
However, revisions were required as followings:
1. Mangroves are important groups of marine plants, together with diverse groups of microorganisms form an ecosystem, featuring various bioactive secondary metabolites. These were missing in this review.
2. It is important to number compounds with Arabic numerals, show the structures of compounds and provide the data of biological activity such as IC50 values. Please provide them in the manuscript.
Others:
1. In the Abstract, authors said this review decried recent discoveries of anti-inflammatory and immunomodulatory marine molecules, which was not exactly consistent with the title.
2. There were too many keywords.
3. What types of organisms did these numbers ‘611.000, 63.900’(P2L49) refer to?
4. According to Chinese characters 食物本草(P6L242), the name ‘Wu shi ben cao’ needs to be revised as ‘Shi wu ben cao’, the alternative name ‘Wu's Materia Medica’ perhaps could be revised as ‘Food Materia Medica’, since ‘食物(Shi wu)’ means ‘Food’.
5. There were many references lack of pages at the end of this manuscript, such as [8], [18], [26], [62], [77], [347].
6. There were a few typo errors, such as Italic font for the genus name ‘Hippocampus’(P3L108&P6L228), and the missing half bracket ‘(Bacteria of the Sea’ → ‘(Bacteria of the Sea)’(P9L345).
Comments on the Quality of English Language
There were a few typo and grammar errors, some of which were given in the comments.
Author Response
The topic of this manuscript is quite interesting, and this review is important. The authors have profound knowledge on the origins of marine pharmacopoeia, which was revealed by the detailed introduction. However, as for the modern era, the review on the immunomodulatory marine compounds was lack of extensive examples and detail information. Of course, this manuscript is worth publishing in this journal.
EAC: Thank you for acknowledging the value of our manuscript. Please find below replies to your comments. We hope we have addressed all your concerns in the newly submitted version of the manuscript.
However, revisions were required as followings:
1 Mangroves are important groups of marine plants, together with diverse groups of microorganisms form an ecosystem, featuring various bioactive secondary metabolites. These were missing in this review.
EAC: Thank you for pointing out this. We have now added a short section (7.3) in which we highlight recent discoveries of lead compounds identified in mangroves habitats. Moreover, elsewhere in the manuscript we have mentioned examples of microorganisms associated with mangroves from which new anti-inflammatory and immunomodulatory compounds have been characterized. Now lines 557 574.
It is important to number compounds with Arabic numerals, show the structures of compounds and provide the data of biological activity such as IC50 values. Please provide them in the manuscript.
EAC: We have included Arabic numerals in the figures describing molecules.
Others:
1 In the Abstract, authors said this review decried recent discoveries of anti-inflammatory and immunomodulatory marine molecules, which was not exactly consistent with the title.
EAC: Thank you for this observation. We have accordingly rephrased the abstract to be consistent with the title, stating that we are providing examples of recently identified marine immunomodulatory lead compounds.
2 There were too many keywords.
EAC: Thank you for this observation. We have now reduced the number of keywords to the following ten: bioprospecting; inflammation; autoimmunity; synthetic biology; drug discovery; genetic engineering; immunomodulation; deep sea; systemic sclerosis; rheumatoid arthritis
3 What types of organisms did these numbers ‘611.000, 63.900’(P2L49) refer to?
EAC: Thank you for pointing out this unclear statement. We have nevertheless rearranged the sentence and the removed these numbers. Now lines 63-67.
4 According to Chinese characters 食物本草(P6L242), the name ‘Wu shi ben cao’ needs to be revised as ‘Shi wu ben cao’, the alternative name ‘Wu's Materia Medica’ perhaps could be revised as ‘Food Materia Medica’, since ‘食物(Shi wu)’ means ‘Food’.
EAC: we are extremely grateful for this observation. We nevertheless removed this name from the manuscript in order to shorten this section.
5 There were many references lack of pages at the end of this manuscript, such as [8], [18], [26], [62], [77], [347].
EAC: thank you, this was fixed.
6 There were a few typo errors, such asItalic font for the genus name ‘Hippocampus’(P3L108&P6L228), and the missing half bracket ‘(Bacteria of the Sea’ → ‘(Bacteria of the Sea)’(P9L345).
EAC: thank you, we have revised these typos. Now lines 101, 192 and 199
Round 2
Reviewer 1 Report
Comments and Suggestions for Authors
I am satisfied with the author's revisions, especially since this article is now more resume and easier to read (although it is still long). I think this review can be accepted now.
Author Response
I am satisfied with the author's revisions, especially since this article is now more resume and easier to read (although it is still long). I think this review can be accepted now.
Thank you!!
Reviewer 2 Report
Comments and Suggestions for Authors
The manuscript has been improved. However, revisions are required again.
1. The review on the immunomodulatory marine compounds need more improvements. Especially, the detail information of compounds includes their names, chemical structures, and the data of biological activity such as IC50 values should be provided, which will attract more attentions from researchers and give them more useful information. For example, ‘diterpenes (C20H32) isolated from the Formosan gorgonians Briareum excavatum and Sinularia querciformis exerted…)’(P13L462). This sentence did not give enough information for the statement.
2. Insufficient information for many references, such as lack of page([73]), or title([76]), were observed again. Although it is not important compared to the whole manuscript, it can affect reviewers in downloading and reading these references and making relevant judgments.
Comments on the Quality of English Language
A few typo and grammar errors were observed, including the same issues as presented in the last round.
Author Response
The manuscript has been improved. However, revisions are required again.
1 The review on the immunomodulatory marine compounds need more improvements. Especially, the detail information of compounds includes their names, chemical structures, and the data of biological activity such as IC50 values should be provided, which will attract more attentions from researchers and give them more useful information. For example, ‘diterpenes (C20H32) isolated from the Formosan gorgonians Briareum excavatum and Sinularia querciformis exerted…)’(P13L462). This sentence did not give enough information for the statement.
Dear reviewer, thank you for your time and precious inputs for improving our manuscript. According to your suggestion, we have now included names and IC50 for all compounds mentioned in our manuscript (highlighted in yellow, when available in the cited references). We agree with you that this information is helpful for the reader.
However, we did not include the structures for all compounds, since we believe this is not within the scope of this manuscript, but also included the structures of the two molecules described in detail in section 8.1 and 8.2. We trust you understand our choice.
2 Insufficient information for many references, such as lack of page([73]), or title([76]), were observed again. Although it is not important compared to the whole manuscript, it can affect reviewers in downloading and reading these references and making relevant judgments.
Thank you for pointing out these issues again. We have now carefully checked the reference list and all typos throughout the text.
Round 3
Reviewer 2 Report
Comments and Suggestions for Authors
The manuscript has been improved.
1. Subscript font for number ‘50’ in the term ‘IC50’.
2. Indeed, there were many references lack of pages were observed. Take the first page of the reference list, [6], [8], [13], [14], [17], and [18] could be easily found without pages.
However, these revisions were not as important as the main text. This manuscript could be accepted.
Author Response
Comment 1. Subscript font for number ‘50’ in the term ‘IC50’.
EAC Response 1: thank you for pointing this out, we have corrected this subscript throughout the text
Comment 2. Indeed, there were many references lack of pages were observed. Take the first page of the reference list, [6], [8], [13], [14], [17], and [18] could be easily found without pages.
However, these revisions were not as important as the main text. This manuscript could be accepted.
EAC Response 2:Thank you. We have checked again these imprecisions.